# Leveraging Stack4Things for Federated Learning in Intelligent Cyber Physical Systems

**Fabrizio De Vita** *,† and **Dario Bruneo** *,†

Department of Engineering, University of Messina, 98166 Messina, Italy
* Correspondence: fdevita@unime.it; (F.D.V.); dbruneo@unime.it (D.B.)
† These authors contributed equally to this work.

**Abstract:** During the last decade, the Internet of Things acted as catalyst for the big data phenomenon. As result, modern edge devices can access a huge amount of data that can be exploited to build useful services. In such a context, artificial intelligence has a key role to develop intelligent systems (e.g., intelligent cyber physical systems) that create a connecting bridge with the physical world. However, as time goes by, machine and deep learning applications are becoming more complex, requiring increasing amounts of data and training time, which makes the use of centralized approaches unsuitable. Federated learning is an emerging paradigm which enables the cooperation of edge devices to learn a shared model (while keeping private their training data), thereby abating the training time. Although federated learning is a promising technique, its implementation is difficult and brings a lot of challenges. In this paper, we present an extension of Stack4Things, a cloud platform developed in our department; leveraging its functionalities, we enabled the deployment of federated learning on edge devices without caring their heterogeneity. Experimental results show a comparison with a centralized approach and demonstrate the effectiveness of the proposed approach in terms of both training time and model accuracy.

**Keywords:** federated learning; intelligent cyber physical systems; stack4things

## 1. Introduction

Since the advent of machine learning, data gathering has always been problematic. Today, thanks to the advancements of ICT technology and the Internet of Things (IoT) spreading evermore, this problem has been largely solved. During the last decade, a wide variety of cyber physical systems (CPS) acting as sensors and actuators have been producing a huge amount of heterogeneous information, actively participating to the big data phenomenon [1]. In such a context, artificial intelligence (AI) plays a key role in enabling the realization of intelligent cyber physical systems (ICPSs), new devices capable of "reason" and making decisions according to the context [2].

As a result of this condition, machine and deep learning applications (e.g., computer vision, speech recognition, natural language processing, etc.) are becoming data hungry, requiring more and more data [3]. However, larger and more complex datasets are also synonyms for more complex and time consuming model training procedures, which can be unaffordable for the hardware of an edge device. A possible solution could be the use of a hybrid approach where the training is done on a powerful machine (e.g., on the cloud) and then performing only the inference process on the ICPS (i.e., the edge), but this could result in poor scalability and bad fault tolerance because of this centralization. Another problem related to this condition is the creation of "data islands" [4], where a constantly increasing huge amount of stored data remains isolated (i.e., not shared with other entities), making more difficult its organization and privacy management.

Federated learning (FL) is an emerging machine learning technique that can address the above-mentioned problems, constituting a new way to perform the training of machine and deep learning models [4] on the edge. The concept at the basis of FL is collaboration (or sharing); given a set of clients (or participants), they are trained on a small part of a dataset under the coordination of a central entity (e.g., the cloud), and share their local models to create a new one synthesizing the "knowledge" of each contributor. Unlike a centralized approach, FL can also preserve the data privacy [5] in a better way. The training process is in fact entirely done on data stored locally on each client, who only share their trained models. In an era where the data contained in devices can be sensitive, the capabilities of this training paradigm to scale while keeping data private enable the realization of more secure applications.

Most of the recent approaches propose the use of FL in smart city scenarios, thereby exploiting the huge number of sensors blended with an environment whose data are generated via ICPS. In such a context, hundreds of clients can participate to the train process with the aim of learning a collaborative global model which is capable of improving its performance on a specific task [6]. Another interesting context where FL could be beneficial is the smart industry; here, fault detection tasks are gaining a lot of interest in order to avoid systems breakdown [7,8]. However, the realization of these techniques (usually based on time series analysis) requires a large amount of sensor data. In this sense considering a scenario made of several industrial plants, they could jointly participate to the FL training process, with the aim of learning a shared model able to detect a wide range of faults in a lower amount of time thanks to the workload split. Of course, the use of FL for time series prediction comes also with several research challenges, which are caused in part by the time dependence between samples, which has to be handled in order to avoid the learning of wrong models.

If on the one hand FL could solve the computing, scalability, and privacy problems; on the other it introduces a set of challenges to be addressed, most of which are related to the use of a distributed approach requiring careful management. Since clients are usually entities distributed over the edge and can be very distant (potentially at the world opposites), problems related to sudden disconnections or limited bandwidth, just to name a few, have to be managed through the realization of effective frameworks. Moreover, if we consider the clients' heterogeneity (both in terms of hardware and software), the implementation is even more challenging.

In this paper, we propose an extension of Stack4Things (S4T), a cloud-based platform developed in our engineering department that allows one to control and orchestrate IoT devices. In a context where the available FL platforms exhibit some limitations, especially in terms of the client addressing and the software they can run, we use S4T to address these problems. Leveraging its functionalities, we implemented a FL framework by adding an AI engine, to enable distributed training over a set of heterogeneous ICPSs without caring about their location, network configuration, and underlying technology (i.e., hardware and software) [9].

The paper's contributions can be summarized as follows: (i) We extended S4T by implementing a set of functionalities that enable this framework to perform a FL approach. (ii) We propose a proof-of-concept (PoC) case study based on smart cities that we think could benefit from using a FL approach. (iii) To prove the effectiveness of the proposed approach, we compared it with a centralized scheme in terms of training time and model accuracy by conducting two sets of experiments.

The rest of the paper is organized as follows. Section 2 reviews the related works in the literature while highlighting the differences from our approach. Section 3 describes in detail the architecture of Stack4Things. Section 4 presents the federated approach we adopted and explains how we implemented it in S4T. Section 5 describes a PoC case study of an urban scenario and presents the experimental results obtained from comparing the proposed approach with a centralized one. Finally, Section 6 concludes the paper and provides the directions for future work.

## 2. Related Works

In this section we report the existing literature putting in evidence the differences in our approach. The work in [4] provides an overview of FL. Here the authors describe different types of possible

architectures with the corresponding advantages and disadvantages, putting also in evidence the challenges of their implementation. In [10] is presented an interesting work which exploits FL for air quality forecasting using unmanned aerial vehicle (UAV) swarms. Thanks to FL's privacy preserving property, this work enables the cooperation of multiple institutions to build a global model while expanding the scope of UAVs. Authors from Google in [11] present a FL approach based on model averaging and produce extensive results demonstrating the effectiveness of this technique when applied to different datasets. The work described in [12] proposes the use of FL for Chinese text-recognition and compares two popular frameworks: TensorFlow Federated (TFF) and PySift (https://www.openmined.org/, accessed October 2020), but as the authors state, these frameworks are still under development. For example, TFF exposes useful interfaces that ease the implementation of federated learning tasks; however, in FL a very important part consists also of the communication between the clients and the cloud; in this sense, TFF does not provide any facilitation to do that. On the other hand, S4T has been designed to establish and manage the communication with clients even if behind a Network Address Translation (NAT) or firewalls [9]. PySift is a FL framework specialized in the use of cryptographic tools to guarantee a secure training process and deployment. The framework, however, allows only the use of Python libraries, such as TensorFlow, Keras, and PyTorch. Unlike PySift, S4T can work also with other machine learning frameworks since its functionalities do not depend on the user code running on top of it. Authors in [13] present a promising framework (called Flower) for the deployment of FL algorithms on heterogeneous clients. Although the framework outperforms others approaches, e.g., TFF and PySift, it can be used to accomplish only FL tasks. In contrast, thanks to S4T modularity, it can be used to manage the training in multiple ways by performing it on the cloud, the edge, or using a FL scheme. In [14], authors propose "In-Edge-AI" a FL framework based on deep reinforcement learning to optimize the computation, caching, and communication in mobile edge computing (MEC). Even if in this work, the authors were able to obtain very good results, the framework is strongly specific in terms of both hardware and software. In this sense, S4T has been designed to work in many different contexts, with a wide variety of clients allowing the implementation of several types of applications on top of it without changing its architecture. Authors in [15] present LEAF, a framework for FL schemes mainly used for benchmarking, which however, is based also in this case on Python simulations. As already said, S4T works independently from the applications deployed on it which can be written in many different programming languages. In [16], authors propose FedSteg, a FL framework for image steganalysis. Even if the obtained results are promising, the authors did not deploy it in a real scenario. In this sense, one of our main contributions is to demonstrate the feasibility of the proposed approach in a real application context. Moreover, in this work the authors focused on FL performance when varying the data payload without providing any insights when varying the number of participants in the training process. The work in [17] describes PerFit a cloud–edge framework that allows one to perform personalized FL schemes in IoT devices using transfer-learning mechanisms. The experimental results section shows how the test accuracy and time cost change according to the number of clients; however, we authors do not make a direct comparison with a centralized approach. On the other hand, our goal in this work is not only to prove the effectiveness of FL, but also to show when this approach should be preferred over the centralized one.

## 3. Stack4Things Architecture

The S4T architecture is split into two parts, namely: the gateway side (i.e., the ICPS) and the cloud side. Figure 1a shows the software architecture on the gateway side where we put in evidence the main component blocks. The core element is represented by the so called lightning-rod engine, which communicates with the cloud through a Web Application Messaging Protocol (https://wamp-proto.org, accessed September 2020) (WAMP) router using a full-duplex channel WebSocket (WS) to send and receive messages from/to the cloud. All aspects related to the cloud communication are implemented through S4T WAMP lib that acts as an interface between the the gateway board and the cloud. Moreover, the lightning-rod implements a plugin loader to remotely inject plugins on the

board in real time. This is done by providing access to the operating system (OS) tools interacting with several OS resources, such as peripherals, filesystem, package manager, and internal daemons. Such a functionality turns out to be very useful, especially for system maintenance and updates.

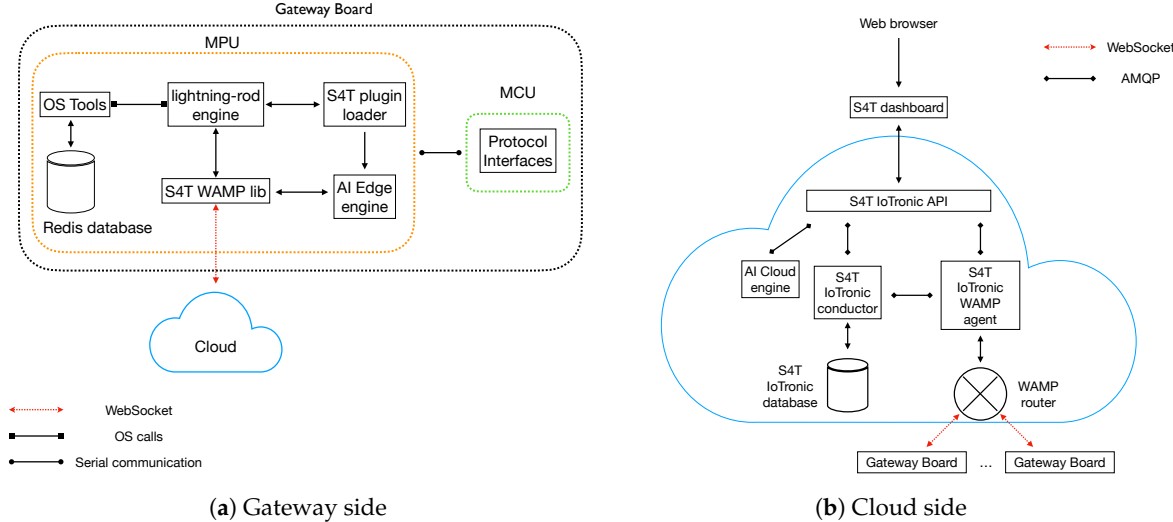

(**a**) Gateway side           (**b**) Cloud side

**Figure 1.** S4T software architecture of the gateway and cloud sides.

The gateway side is also provided with an AI edge engine to perform the training and the inference processes directly on board; such a functionality has been introduced in another work [18] where we extended S4T to work in an industrial scenario; however, in the previous version of the engine, it was only able to perform the inference process. In this new version, we introduced also the possibility to train a model in order to enable the FL scheme inside S4T. In particular, the engine is able to retrieve data coming from the cloud (via WS) and the Micro Controlling Unit (MCU) via several protocol interfaces (Bluetooth Low Energy (BLE), Wi-Fi, CAN, MODBUS, PROFINET, etc.). The Micro Processing Unit (MPU) and MCU communications are implemented using a shared key/value database, called Redis. Specifically, when a new chunk of data arrives at the gateway board MCU, it uses the serial communication with the MPU to write the new measurement in the Redis database by setting as key the name of the sensed data (e.g., the sensor name) and as value the actual measurement. On the other hand, the MPU accesses the database using OS Tools and forwards the data to the AI edge engine to perform the training or inference process. Finally, when the data come from the cloud, they are simply forwarded to the engine via S4T WAMP lib.

With respect to the cloud side, Figure 1b depicts its architecture. Similarly to the gateway side, in this case also, we identify the core element of the architecture represented by the IoTronic component that can be considered as the cloud counterpart of the aforementioned lightning-rod engine. IoTronic consists of three main elements, namely: S4T IoTronic WAMP agent, S4T IoTronic conductor, and S4T IoTronic database.

When a request is sent/received from/to the cloud, the WAMP router interacts with S4T IoTronic WAMP agent that converts Advanced Messages Queueing Protocol (https://www.amqp.org, accessed September 2020) (AMQP) messages into WAMP messages (and vice versa). Thus, the agent acts as the bridge between the cloud and the boards. The choice of AMQP is justified because S4T is an OpenStack extension, and for better integration with the rest of the framework, its components should "communicate" using this protocol. Moreover, AMQP is provided with a set of functionalities that ensure its reliability, security, and scalability due to the possibility of its use as a point-to-point or publish/subscribe protocol [9]. In general, the use of OpenStack as an open source cloud platform fits well with the needs of a FL framework application, such as storage, computation, and network management.

The S4T IoTronic "conductor" acts as a controller to perform read/write operations on the S4T IoTronic database, by executing a set of remote procedure calls (RPCs). This design choice has several advantages, such as security (which is a key aspect for an industrial application), as the presence of an intermediate entity allows one to avoid potentially dangerous operations on the database. The S4T IoTronic database stores all the information related to the registered boards (board identifiers, applications port numbers, the type of service they are running, hardware and software characteristics, etc.). Specifically, it keeps track of the connected boards and dispatches the RPCs, among other components.

Like the gateway, also the cloud is provided with an AI engine to perform both the training and the inference processes. The engine communicates with the rest of the system via S4T IoTronic API, enabling it to perform the training, the inference, and the deployment of machine learning models to the gateway boards. Specifically, when a new trained model is ready to be injected to the edge, the AI engine sends an AMQP message (containing the trained model) to the S4T IoTronic WAMP agent using the S4T IoTronic API. The trained model is then injected to the gateway boards through S4T plugin loader and passed to the AI edge engine. Finally, the S4T dashboard is the entry point that allows one to access all the S4T functionalities using the S4T IoTronic API via a Web browser.

## 4. Federated Approach

In the previous section, we explained in detail how the S4T architecture is organized and how its internal components interact. In this section, we describe how we implemented the FL approach while exploiting the functionalities of our platform. FL is an emerging machine learning technique where a group of participants collaborate to train a shared deep learning model without revealing their data in order to reduce the overall training time while preserving data privacy [19,20]. In such a context, we can identify two entities, namely: the client and the server. With respect to the client (traditionally a passive element for these kinds of applications), it becomes a key component of the entire system architecture actively participating in the "knowledge building" process and performing the actual training of its local model. On the other hand, the server plays a much more complicated role, since it has to coordinate a large number of clients other than aggregate the local models of each participant into a shared one [13].

The motivation of FL comes from the necessity to implement a new way to perform the training process on large datasets. By exploiting the distributed nature of this approach, it is possible to improve the overall system scalability and fault tolerance while preserving privacy. Figure 2 shows a possible FL scheme where the data stored inside each client are used to train the local model. Moreover, the data are not shared either with other clients or with the cloud, thereby preserving the privacy.

In such a context, the FL algorithm consists of the iteration of four steps: (i) global model distribution, (ii) local training, (iii) weights aggregation, and (iv) global model update. During the first step, the central entity (i.e., the server) contacts each participant and sends the global model weights (if this is the first training round the weights should be considered initialized to random values). On the client side, when the global model is received, it performs a fixed number of training steps (usually agreed with the server) and sends it back to the server. Once each client returns the trained model to the server, it executes a weights aggregation procedure to merge these models into a global one condensing the information coming from each client. In this sense, FL proposes several aggregation strategies however most of them consist in using the FederatedAveraging (FedAvg) algorithm [13] where the clients weights are averaged according to the following equation:

$$W_g^L = \frac{1}{N_c} \cdot \sum_{c=1}^{N_c} W_c^L, \tag{1}$$

where $W_g^L$ and $W_c^L$ are the weight matrices of the $L$-th layer of the global and client models respectively, and $N_c$ is the the number of clients participating in the training. A possible implementation of FedAvg is shown in Algorithm 1. The first part of the algorithm is used for the parameter initialization. A global

model is created with random weights (line 1), and the numbers of training epochs $E$, clients $N_c$, and training rounds $T_r$ are set (lines 2–4). The rest of the algorithm is the core of FL. The cloud sends in parallel the global model to every client $c$ that has been selected to participate in the training process (line 7). Then, each client performs (in parallel with the others) a fixed number of $T_r$ optimization steps using its local data (line 9). When the clients finish the training process, the cloud waits for the models (line 12) necessary to perform the aggregation. In such a context, we implemented a timeout mechanism in order to avoid long time bottlenecks caused by very slow or faulty clients that would negatively affect the performance of the entire training process [21]. In particular, when the cloud waits for the clients models, it triggers a countdown, after which the connections with clients not satisfying this time requirement are cut. By doing so, we are able to mitigate the bottleneck problem, while also solving the issue related to the clients failure that would cause an "infinite" waiting of the cloud side. After this step, the cloud aggregates the models (line 13) and finally updates the global model (line 14). Steps from line 6 to 14 are repeated until the number of training epochs $E$ is reached.

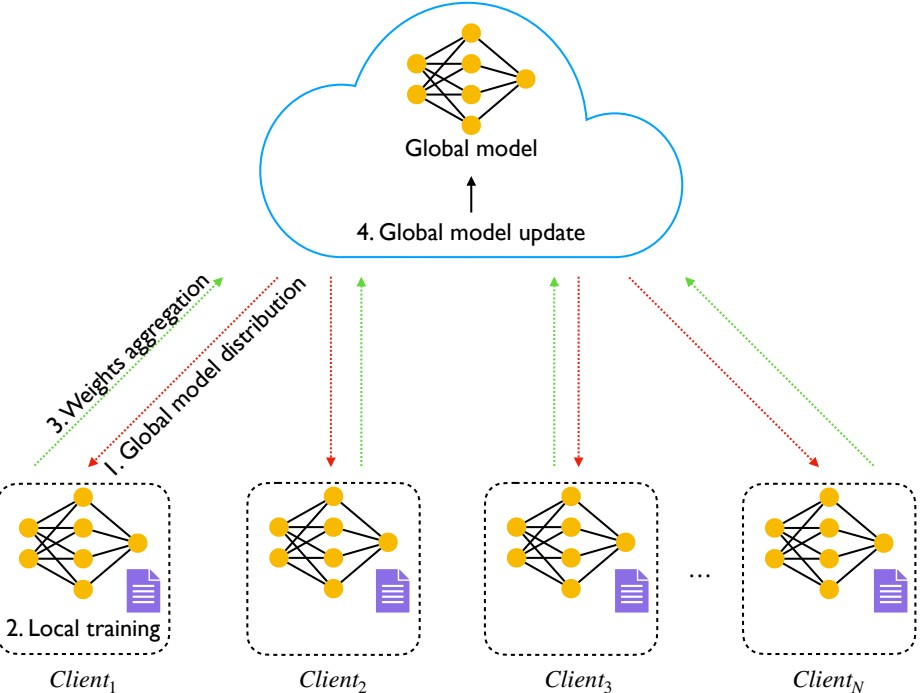

**Figure 2.** Deployment of a FL scheme.

Due to the large number of components and system complexity, for a better understanding Figure 3 reports the entire workflow of the FL approach implemented in S4T for each client.

The communication scheme between the client and the cloud can be split into two phases, namely: registration and federated learning. With respect to the registration part, assuming a condition where S4T is installed from scratch, the very first step is to register the client to the cloud in order to access the functionalities of the framework. The registration is done by making a request via WebSocket using the lightning-rod engine (step 1). On the cloud side, the request is processed by the WAMP router and forwarded to the S4T IoTronic WAMP agent. As mentioned, the task of the agent is to convert the message request in an AMQP format, making it readable for every OpenStack component. After this translation process, the agent uses AMQP to contact the S4T IoTronic conductor, which takes charge of the request, and forwards it to the S4T IoTronic database using a suitable SQL query (step 2). At the end of this procedure, the database contains an entry of the registered board (step 3) (that will persist as long as a de-registration request is performed) and a message confirming the registration in sent to the client gateway board (step 4).

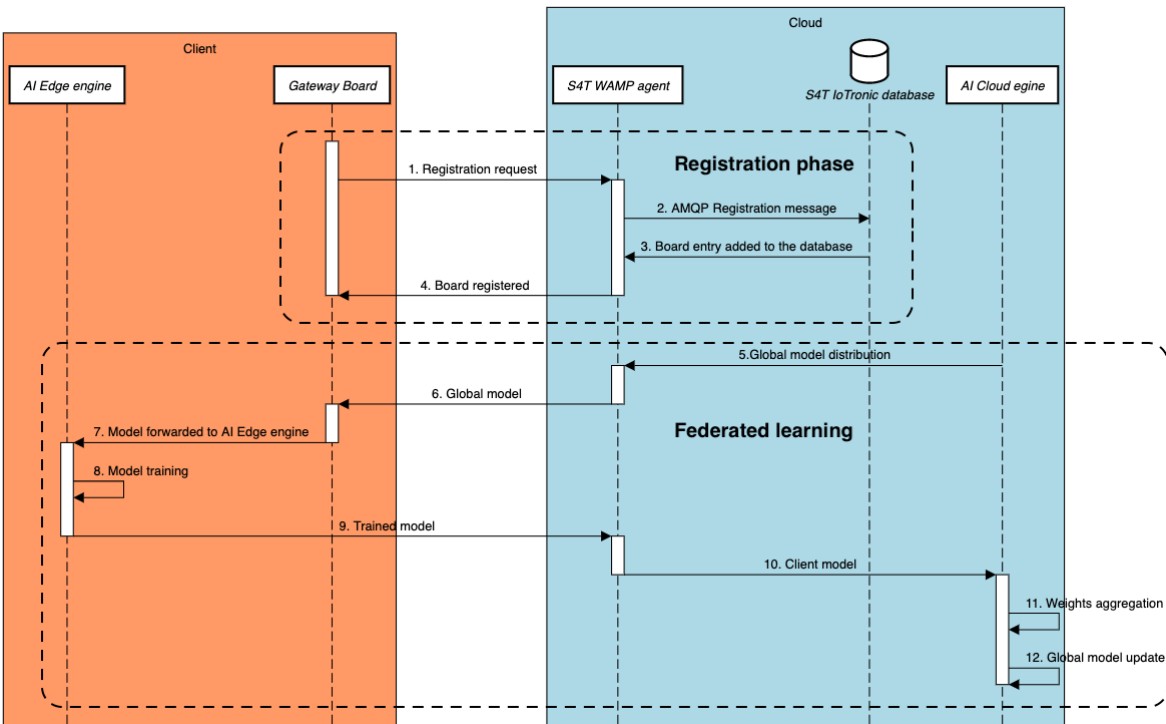

**Figure 3.** Client registration and FL training in S4T.

The second part of the workflow consists of the actual FL algorithm, whose steps are iteratively repeated until a fixed number of training epochs is reached (as shown in Algorithm 1). Once each client confirmed its participation to the training process, the cloud sends to each one of them an initial global model whose weights are randomly set. In S4T this is accomplished by the AI cloud engine which forwards the global model to the S4T WAMP agent (step 5) that is responsible for distributing it in parallel to all the clients (step 6). When the client receives the model, it is passed to the AI edge engine (step 7) for the training (step 8). Then, the client-trained model is sent back to the S4T WAMP agent (step 9) and passed to the AI cloud engine (step 10). After receiving a model from each client, the engine performs the weights aggregation function to merge them (step 11) and the global model is updated (step 12).

---

**Algorithm 1:** *FedAVG* algorithm.

---

 1　build a global model and initializes the weights $W_g$ to random values
 2　set the number of training epochs $E$
 3　set the number of clients $N_c$ that will participate to the training
 4　set the number of training rounds $T_r$ to be executed on the client side
 5　**for** epoch = 1 to $E$:
 6　　**do in parallel**
 7　　send global model to each client $c$ participating to the training
 8　　**for** round = 1 to $T_r$:
 9　　　update local model weights $W_c$
10　　**end for**
11　　**end**
12　　receive client models
13　　aggregate the weights using Equation (1)
14　　update global model weights $W_g$
15　**end for**

---

## 5. Experimental Results

In this section, we report the experimental results from testing the proposed FL approach while exploiting the S4T functionalities. In particular, our goal is to show the feasibility of our system while demonstrating the effectiveness of the FL approach when compared to a centralized one. With respect to the experimental setup, we constructed a distributed scenario where the cloud instance of S4T is deployed on a public server available in our university department. For the client side, in order to put in evidence the capability of the framework to work with heterogeneous hardware and software, we considered several clients: a laptop, three Raspberry Pi 3s, and a NVIDIA Jetson Nano, located in different buildings of the university campus such that they did not lay on the same network, thereby emulating a distributed deployment, as shown in Figure 4. Given the above-described experimental setting, we selected a hardware setup to best emulate a realistic FL scenario. Moreover, if we consider an application context where the clients are represented by organizations, it is quite common to have a number of participants compatible with the one proposed in this work. In Table 1 we report the hardware configurations of clients and server in terms of CPU, GPU, and RAM.

**Table 1.** Server and client hardware configurations.

| Hardware Configurations | | | |
|---|---|---|---|
| Hardware | CPU | GPU | RAM |
| Server | Intel Xeon @ 2.13 GHz | - | 16 GB DDR4 @ 2667 Mhz |
| Raspberry Pi 3 | ARM A53 @ 1.4 GHz | - | 1 GB LPDDR2 |
| Jetson Nano | ARM A57 @ 1.43 GHz | NVIDIA Maxwell @ 921 MHz | 4 GB LPDDR4 @ 1600 MHz |
| Laptop | Intel Core i7 @ 2.6 GHz | 4 GB ATI Radeon Pro | 16 GB DDR4 @ 2667 Mhz |

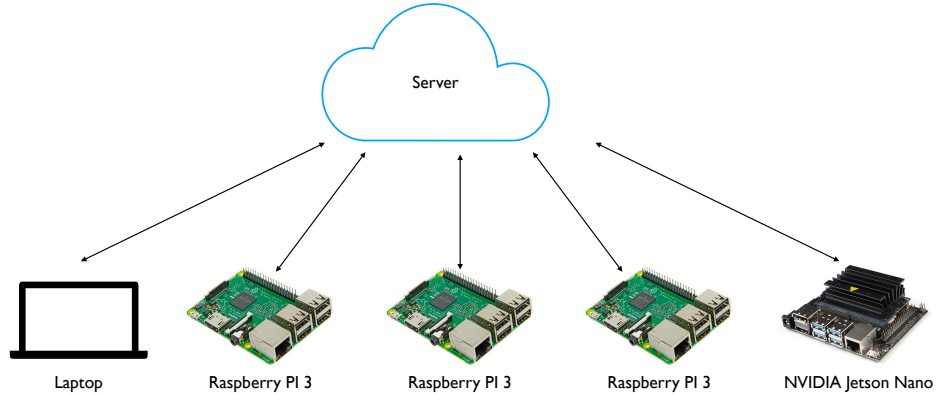

**Figure 4.** Federated learning training scenario consisting of five clients.

We evaluated a PoC case study which can benefit from the use of a FL approach. Specifically, we chose a smart city scenario where a set of traffic cameras was deployed at road junctions (as shown in Figure 5) collecting images of different types of vehicles (cars, trucks, buses, etc.). Each camera was responsible for monitoring only a part of the road junction and was equipped with the hardware that enabled it to do its computations and connect to the Internet; however, for privacy reasons the data stored by the camera were private and could not be shared with other entities. Moreover, each camera had its own specific framing and settings (i.e., the positioning, the exposure parameter, the shutter length, etc.), which make the captured images very different from the others. In this scenario, we assumed to have a S4T deployment where the cameras act as clients communicating with the cloud using the mechanisms we described in Section 3. In such a context, we propose a smart city application which exploits deep learning techniques to classify the vehicles crossing the road junction for traffic analysis purposes. To this aim, traffic cameras can use their data to realize a model capable recognizing which types of vehicles are crossing the road. However, it is known that computer vision

applications usually require a huge amount of data and much training time to reach satisfying levels of accuracy. Since cameras images are very different, the training process strongly depends on their local data, which makes the use of more sources of information a key element to improve the overall performance of the system. In this sense, such an application could exploit a FL approach where several traffic cameras cooperate, sharing only their local models to create a cooperative (or global) one which synthesizes the useful data information of each camera while abating the training time and conserving the privacy. Moreover, the use of FL enables the implementation of a continuous learning mechanism where clients are periodically updated according to the available data; however, the realization of this technique can be challenging when the amount of training data is low. In this sense, FL provides "virtual" access to a large number of data samples coming from several sets of sources that can be exploited to continuously improve the performance of a task.

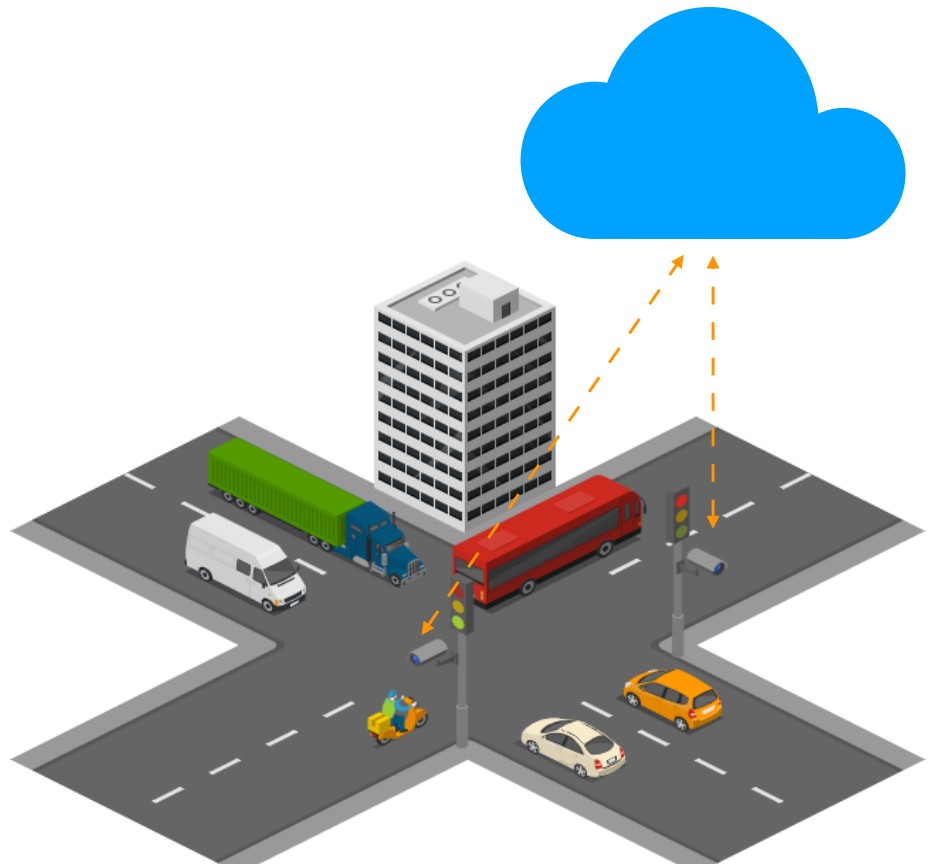

**Figure 5.** Smart city scenario.

To measure the performance of the proposed FL framework both in terms of training time and model accuracy, we compared it with a centralized approach. The above-described urban scenario has been simulated using the Miovision traffic camera dataset (MIO-TCD), the largest dataset for traffic analysis containing images of surveillance cameras deployed in America and Canada [22]. The dataset consists of 786, 702 labeled images belonging to 11 classes, i.e., articulated truck, background, bicycle, bus, car, motorcycle, non-motorized vehicle, pedestrian, pickup truck, single unit truck, and work van. Since the goal of our PoC was to demonstrate the effectiveness of FL when working with heterogeneous devices that can have a low computing power, we selected a subset of four classes (i.e., articulated truck, bus, car, and motorcycle) representative of the most popular vehicles present in an urban scenario. Even if we reduced the number of classes, we maintained a large number of samples (about 30 K) to emulate a real FL scenario where the workload split on a huge dataset can improve the training performance. Of course, considering a real application scenario, the labels are

not known a priori; in this sense, it is possible to use automatic data labeling tools that allow one to create labeled datasets to be used in supervised approaches. To prove the effectiveness of the proposed approach, we conducted two set of experiments: in the first one we considered to have a variable number of labeled samples for each of the above selected classes, in order to obtain different training conditions. In the second case, we considered to have a variable number of clients while keeping fixed the dataset size, to observe how the FL reacts when changing the number of participants in the training process. Moreover, in both the experiments we considered a test set with a total of 800 samples (200 for each class) to evaluate the performances of the two approaches (i.e., centralized and federated).

The problem of classifying vehicles that we mentioned when describing the application scenario is that it has been tackled as a supervised deep learning approach using a convolutional neural network (CNN) (shown in Figure 6), which is particularly suitable for tasks of this type. Table 2 shows the hyperparameter settings we adopted for our deep learning model. The CNN was composed of 7 layers where the first four consisted in the alternation of convolutional layers with 32 filters; a kernel size set to 3; strides of 4 and 1 respectively; and max pooling layers with a pool size of 2. The rest of the CNN uses 3 fully connected layers with 512, 128, and 4 neurons respectively to perform the actual image classification. For each layer, we adopted a rectifier linear unit (ReLU) activation function representing nowadays a sort of standard for these neural network architectures. To avoid network overfitting, we used a dropout technique with a drop rate of 0.2 for the first fully connected layer and 0.5 for the others. Finally, we used the Adam optimizer with a learning rate of 0.001 and set the maximum number of training epochs to 100.

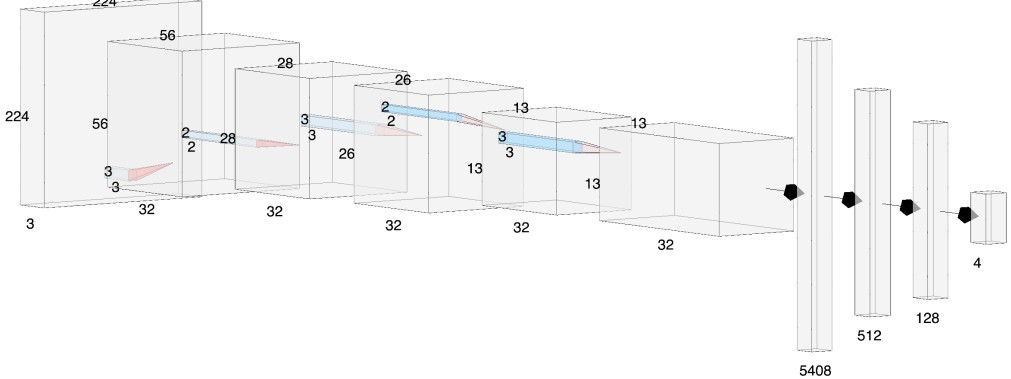

**Figure 6.** CNN architecture.

**Table 2.** CNN parameters.

| CNN Model Architecture | | | |
|---|---|---|---|
| No. of layers | 7 | Optimizer | Adam |
| Activation function | ReLU | Learning rate | 0.001 |
| Pooling method | Max-pooling | Dropout rate | [0.2, 0.5] |
| Kernel size | 3 | Max training epochs | 100 |

Figure 7 depicts the loss of the federated model when varying the number of samples in the dataset and computed as the mean of the losses of each of the five clients' local models:

$$-\frac{1}{N_c} \cdot \sum_{c=1}^{N_c} \sum_{i=1}^{M} \sum_{j=1}^{C} y_{i,j} \cdot log(\hat{y}_{i,j}), \tag{2}$$

where $M$ is the number of training samples, $C$ is the number of classes, $y_{i,j}$ is the ground truth, and $\hat{y}_{i,j}$ is the value predicted by the CNN model.

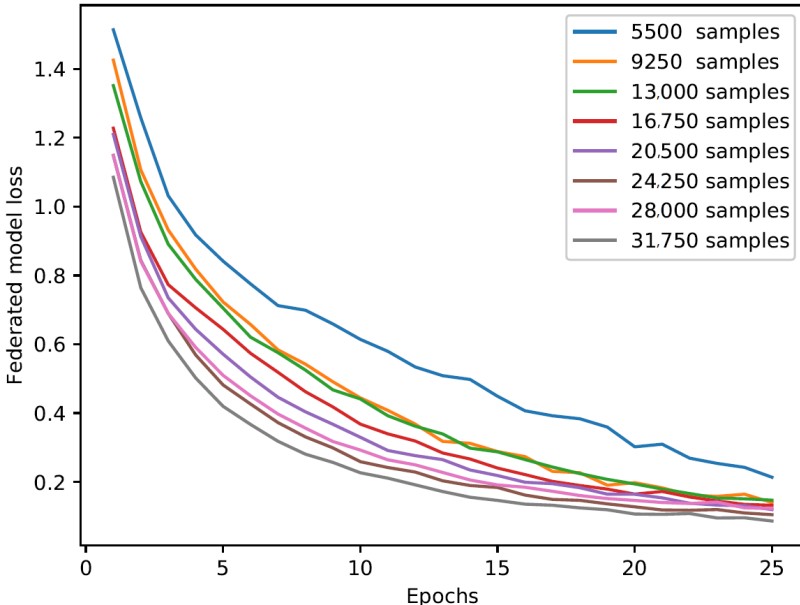

**Figure 7.** Loss of the federated model loss when varying the number of samples in the dataset.

In general, we can observe that the FL approach was successful for the model training; in fact, in all cases the losses exhibit a decreasing trend which reaches his lowest value as the number of samples in the dataset increases. Moreover, we notice that the number of training epochs (originally set to 100) stops at 25; this is due to an early stopping technique applied to the loss function that stops the training in advance if the model is not able to improve (i.e., the loss function does not decrease), thereby avoiding model overtraining that can lead to overfitting.

Figure 8 shows two comparisons we made considering a centralized approach and a federated one using the five clients we introduced at the beginning of this section (see Figure 4 and Table 1). Figure 8a depicts a comparison in terms of training time when varying the number of samples. As expected, in both cases the training time increases almost linearly as the number of training samples increases; however, the federated approach is able to significantly reduce it thanks to its distributed nature that allows one to train in parallel a large number clients, thereby splitting the workload. From the plot we can also notice that the time difference between the approaches tends to become larger as the samples increase in number, thereby suggesting that the effectiveness of the FL method raises accordingly. In particular, considering 5.5 K samples (i.e., 1.1 K for each client), the FL and centralized approaches registered training times of 406 and 470 s respectively, thereby resulting in a reduction of about 13.5%. On the other hand, considering 31.75 K samples (i.e., 6350 for each client), we obtained a training time of about 1769 s for the FL and 2719 s for the centralized approach with a time reduction of about 35%. For a better visualization in Figure 9a, we report the FL speed-up in terms of training time with respect to the centralized approach, when varying the number of samples. In general, the speed-up increases accordingly with the number of samples, thereby demonstrating the effectiveness of FL in reducing the training time, as already depicted in Figure 8a.

Regarding Figure 8b, it represents a comparison between the above-mentioned approaches in terms of accuracy when varying the number of training samples. In such a context, we can observe that as long as the samples increase the accuracy, the accuracy from the test set increases accordingly in both cases. In particular, the federated model reaches an accuracy of 94.375% (in the best case) which is perfectly comparable with the one reached by the centralized model equal to 94.625%. In general, the accuracy of FL tends to approach the centralized one as the dataset becomes larger reaching in some cases (i.e., for 13 K and 16.75 K samples) the exact same values. Such a result further demonstrates the effectiveness of the FL approach, which is able to strongly reduce the training time while maintaining almost the same accuracy performance with respect to a centralized model. In such

a context, we obtained an accuracy of 84.875% for 5.5 K samples, reaching the highest value of 94.375% with 31.75 K samples, thereby registering an overall improvement of about 11%. Figure 9b depicts the FL accuracy loss percentage with respect to the centralized approach. In general, the trend tends to decrease, demonstrating the FL capability to reach a good level of performance approaching the centralized accuracy level.

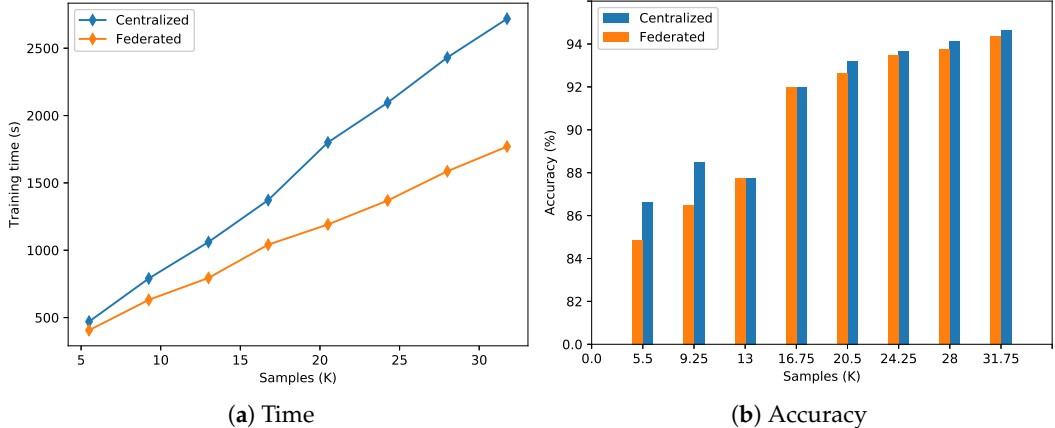

**(a)** Time            **(b)** Accuracy

**Figure 8.** Time and accuracy comparison between a centralized approach (blue) and a federated one (orange) when varying the number of samples in the dataset.

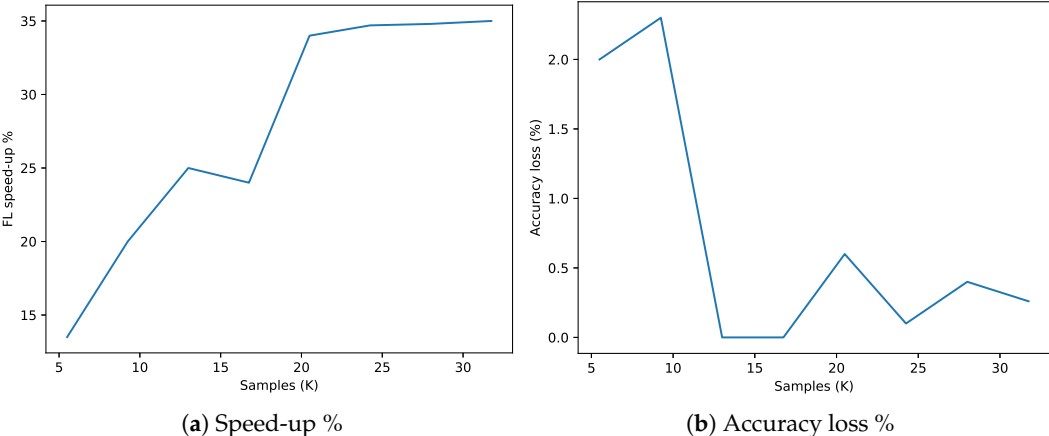

**(a)** Speed-up %            **(b)** Accuracy loss %

**Figure 9.** FL speed-up and accuracy loss with respect to the centralized approach when varying the number of samples in the dataset.

Figure 10 depicts a comparison between the two approaches when varying the number of clients participating in the training process and keeping fixed the dataset size (i.e., 31.75 K samples) for the centralized approach, while for the FL, we gradually increased the dataset size by 6.35 thousand according to the number of clients participating in the training process, such that for five clients the training samples used in both the approaches coincide (i.e., 5 × 6.35 K = 31.75 K). Figure 10a represents a time comparison when varying the number of clients from two to five. The plot shows that as the number of clients increases, the overall FL training time increases accordingly, while always remaining below that of the centralized one. Such a result should not be misleading; in fact, if on the one hand the increment of the number of clients allows one to better distribute the workload; on the other it also introduces an overhead due to the communication and coordination with the server, aside from a higher time to aggregate the models weights, which results in an increment of the training time. Another aspect that should be taken into account is related to the clients' heterogeneity. In particular, when working with FL approaches, a very challenging problem is the bottleneck caused by the slowest client in performing the training process [23]. As for the centralized nature of the

aggregation process, it requires the models of every client participating to the training; as a result, the overall performance of FL is heavily affected by the hardware of the clients. A possible solution to reduce this effect could be the use of timeout mechanisms during the clients' training in order to cut those connections that would cause the inevitable bottleneck (as we did, when explaining the proposed algorithm). Another promising approach involves quantization or compression techniques to reduce the complexity of the deep learning models, thereby making their training process suitable, even for those clients with hardware constraints, and reducing the bottleneck time [24].

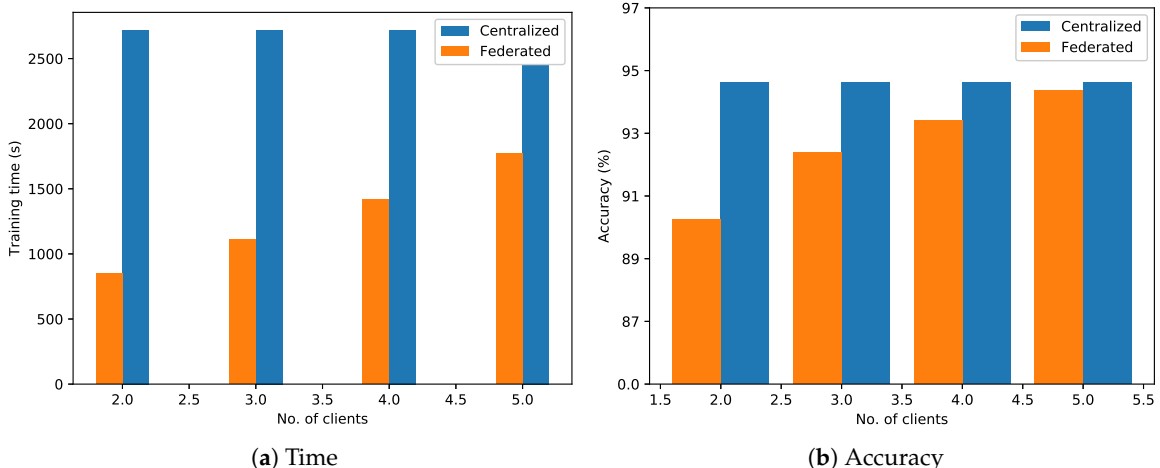

**Figure 10.** Time and accuracy comparison between a centralized approach (blue) and a federated one (orange) when varying the number of clients.

Figure 10b shows a comparison in terms of accuracy when increasing the number of clients. As one could expect, the model accuracy increases accordingly with the number of participants, and this happens because the increment of participants during the FL process is equivalent to "virtually" adding more data for the training process. In fact, even if the clients do not actually share the local data, their trained models are affected by the data; in this sense, when the cloud does the aggregation process, the cooperative model reflects the contribution of all the participants' data. As a result of this condition, the accuracy of the federated model increases accordingly with the number of clients, starting with an initial value of 90.25% and ending with an accuracy of 94.375%, which ends up also being the closest to the one reached by the centralized approach (i.e., 94.625%). With the aim of computing the FL speed-up and accuracy loss when varying the number of clients, we conducted a new set of experiments where we fixed the dataset size to 31.75 K samples also for the FL approach. The experimental results reported in Figure 11 show that the FL speed-up tends to increase according to the number of clients thanks to the split of the workload that allows one to train each client on a lower number of samples, thereby reducing the overall training time (see Figure 11). With respect to the accuracy loss, we can observe a minimum change of the percentage which remains bounded between 0.26% and 0.27%, confirming once again the capability of FL to maintain an accuracy which is perfectly comparable with one obtained using a centralized approach.

The obtained results from these experiments put in evidence the cases in which the use of FL can be beneficial. According to the plots shown in Figure 8, when the number of samples is low, the FL loses its effectiveness in terms of both time and accuracy. However, the larger the amount of data available, the more convenient the use of a federated approach, which allows one to strongly reduce the training time while maintaining almost the same performance in terms of accuracy. With respect to the number of clients, its increment causes an overall growth of the training time due to the introduction of an overhead necessary to coordinate a larger number of entities (see Figure 10). In this sense, FL should be seen not as a replacement for more "traditional" techniques, but as a valid alternative when specific requirements in terms of hardware, software, and application context are satisfied. Nevertheless,

the training time always remains below the one obtained from using a centralized approach; moreover, as the number of contributors to the FL increases, this implies also the creation of a shared model which is affected by a larger amount of data with a consequent increment of the overall accuracy.

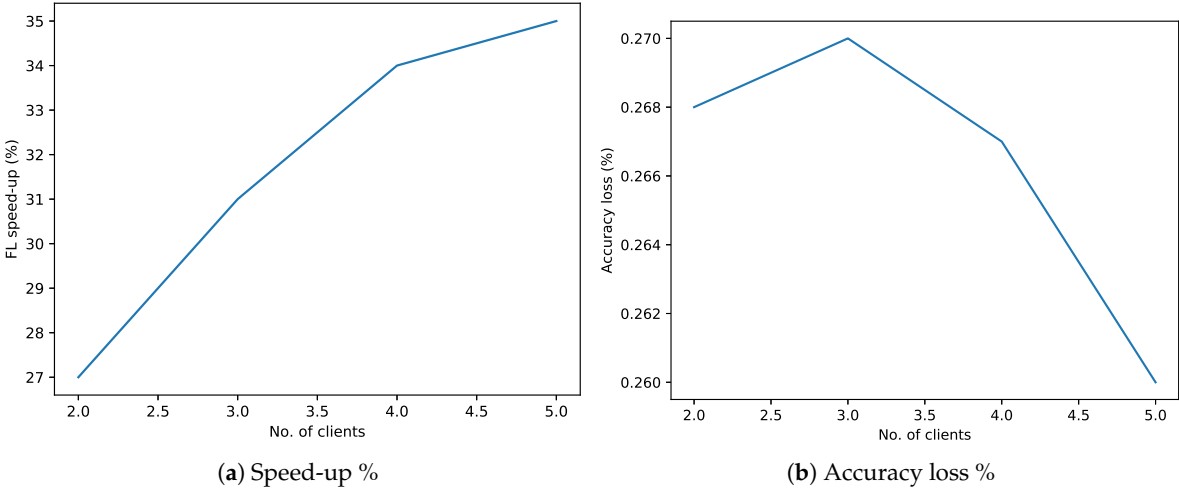

(**a**) Speed-up %
(**b**) Accuracy loss %

**Figure 11.** FL speed-up and accuracy loss with respect to the centralized approach when varying the number of clients

## 6. Conclusions

In this paper, we presented an extension for the execution of FL on S4T a cloud platform we developed in our department that allows one to manage fleets of heterogeneous ICPSs without caring for their network configurations. We presented a PoC case study analyzing a smart city urban scenario, wherein we think that the use of a FL can be beneficial. To measure the performance of the proposed solution in terms of training time and accuracy, we conducted two sets of experiments comparing a centralized approach with a federated one when varying the number of samples in the dataset and the number of clients (or participants) in the training phase. Experimental results demonstrate that the proposed FL technique is able to reduce the training time when compared with a centralized model, and the amount of saved time increases accordingly with the number of samples in the dataset while maintaining almost the same accuracy as the other approach. The use of more clients allows one to better split the workload; however, this also results in the introduction of an overhead caused by a larger number of exchanged messages to coordinate the participants. Despite this overhead, the training time in a federated scenario is still lower than the time required in centralized deployment; moreover, the use of more participants implies also the "virtual" availability of more training data, resulting in an improved global model in terms of accuracy. In an era where smart services are becoming more resource and data demanding, while imposing strict requirements (e.g., latency and privacy) that the cloud is unable to meet, FL is emerging as a new paradigm that addresses these challenges. For example, if we consider a 5G scenario where the European Telecommunications Standards Institute (ETSI) MEC is one of the core elements at the basis of this technology, the use of a distributed approach can be considered a promising solution to build new types of deep learning applications where user equipment (UE) cooperates together to perform a common task. In this sense, MEC can act as a central entity capable of delivering cooperative models among the users. S4T is based on the open-source platform OpenStack which represents the de facto standard of the cloud platforms. Thanks to its modularity, service interoperability could be implemented with commercial frameworks (e.g., Amazon AWS), or other OpenStack modules such as StarlingX (https://docs.starlingx.io/, accessed December 2020) for the deployment of ultra-low latency applications on the edge. In such a context, AI can improve the quality of smart services and should encourage the investigation of new techniques as a foundation of the next generation of mobile cellular networks. Future works will be

devoted to the implementation of more sophisticated FL algorithms, to the realization of a deployment with a larger number of clients, and to the realization of a real smart city application.

**Author Contributions:** Conceptualization, F.D.V., and D.B.; software, F.D.V.; validation F.D.V. and D.B.; investigation, F.D.V. and D.B.; writing–review and editing, F.D.V. and D.B.; supervision, D.B. All authors have read and agreed to the published version of the manuscript.

**Funding:** This research received no external funding.

**Conflicts of Interest:** The authors declare no conflict of interest.

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
