# Peer review of "Leveraging Stack4Things for Federated Learning in Intelligent Cyber Physical Systems"

_jsan, doi:10.3390/jsan9040059_

Round 1

Reviewer 1 Report

The article refers to up-to-date problems of Industry 4.0 applications in real world. It presents the CPS problems related to data acquisition and data analysis. Machine Learning (ML) tools are take into consideration and discussed. As a solution Federated Learning (FL) is proposed. Presented experimental results show a comparison with a centralized approach and demonstrate the effectiveness of the proposed approach both in terms of training time and model accuracy.

The title of the manuscript corresponds with its content and is properly worded. The article is written in a very good form. Its structure is clear with an intriguing formula for presenting a research problem and its solution. Also the terminology used is accurate and compliant with the current standards. The article presents new and original results, achieved with the use of well-designed research. These results are discussed and presented clearly, also with the use of appropriate graphics. The authors have stated clearly what they have achieved and why it has been necessary to undertake the analysed subject. As a results of all this, the article presents of vital importance to the development of knowledge in Industry 4.0 research challenges.

The subject of the study is up to date and it has been almost appropriately substantiated with reference to the present state of knowledge. However the analysis, proposed solution and its background have a single weakness. The FL is proposed as an alternative ML solution. The Authors show many advantages of this idea, also referring to several already published articles (very good selection of references, most of them were published in 2018-2020). Unfortunately they do not compare FL with other ML methods for time series analysis (presenting its advantages and disadvantages), for example SVM. In my opinion, to make this article an excellent this should be discussed too (especially in chapter 1 and chapter 2). For  this purpose I recommend the Authors such references as:

  1. Kozłowski E., Mazurkiewicz D., Żabiński T., Prucnal S., Sęp J. - Machining sensor data management for operation-level predictive model. Expert Systems with Applications 2020; 159: 1-22, https://doi.org/10.1016/j.eswa.2020.113600.
  2. Manco, G., Ritacco, E., Rullo, P., Gallucci, L., Astill, W., Kimber, D., & Antonelli, M. Fault detection and explanation through big data analysis on sensor streams. Expert Systems with Applications 2017; 87, 141–156. https://doi.org/10.1016/j.eswa.2017.05.079.

Author Response

A1.1: Thank you for your suggestion. Indeed, the use of Federated Learning for time series analysis is gaining a lot of interest in the smart industry contexts  where time series analysis techniques are typically adopted to perform fault detection tasks. In particular, using this approach several industrial plants could cooperate (without sharing their data) for the realization of collaborative models that can abate the learning times and improve the overall performance of the detection task. However, the use of Federated Learning for time series analysis is still challenging to implement because of the time dependence between samples and requires careful management to avoid wrong results. In the revised version of the manuscript we discussed these aspects that can be found highlighted in blue in lines (47-55) at page 2. We also added the suggested references to improve the overall quality of the manuscript. 

Reviewer 2 Report

The authors proposed a federated learning approach to enhance the efficiency of machine learning algorithm training. The federated learning de-centralizes the computation and time requirements to train the ML models which enables speed up and enhances privacy. This approach enhances big data privacy since clients only train on parts of the data.

The proposed approach has advantages over centralized learning. The concept and direction of this research are interesting. The manuscript misses a few key points.

  • One of the major drawbacks of the contribution is not discussing technical results in terms of percent speedup and accuracy.
    1. The raw numbers provided do not provide a complete understanding of the limitation and advantages of the proposed work. When does the speed up saturate (if any), how much % is the speedup, how much % is the accuracy (compared to centralized approach), etc.? I believe the 90.25% and 94.375% are about the accuracy of the FL only; not clear.
    2. The accuracy between federated and centralized approaches is very close. How can this be justified when the overhead and complexity of the federated approach is considered?
    3. Varying the number of clients from 2 to 5 does not provide enough details with respect to big data. The motivation is that big data is getting overwhelmingly large, yet the analysis is done with up to 5 clients. A more structured experiment is needed to evaluate the outcomes of FL (in this case the impact of number of clients on accuracy)
  • Proofreading is required. There are several sentences that are hard to read or could be written better. For instance, “actuators has produced (and still produce)” can be replaced with “have been producing”, “making more difficult its organization”, etc.

Author Response

Q2.1

One of the major drawbacks of the contribution is not discussing technical results in terms of percent speedup and accuracy. The raw numbers provided do not provide a complete understanding of the limitation and advantages of the proposed work. When does the speed up saturate (if any), how much % is the speedup, how much % is the accuracy (compared to centralized approach), etc.? I believe the 90.25% and 94.375% are about the accuracy of the FL only; not clear.

A2.1:

Thank you for the suggestion. In our specific experiments we did not find a saturation speed in either of the two experiments we conducted. In this sense, this is strictly related to the hardware, the number of clients, and the application scenario taken into consideration. For a better understanding of the results,  in the revised version of the manuscript we added the speed and accuracy improvements when varying the data size and the number of clients respectively.  Considering the first experiment scenario where the number of clients is fixed to five and the dataset size changes from 5.5 K (i.e., 1.1 K for each client) to 31.75 K (i.e., 6.35 K for each client), we registered a reduction of the training time of about 13.5% in the smallest case up to 35% considering the largest one. In terms of accuracy (assuming the above mentioned scenario), it improves accordingly with the training data. In such a context, we obtained an accuracy of 84.875% up to 94.375% with an overall improvement of about 11%. In general, the speed-up increases accordingly with the number of samples, thus demonstrating the effectiveness of FL in reducing the training time. In terms of accuracy loss with respect to the centralized approach,  FL exhibited a decreasing trend that demonstrates the FL capability to reach a good level of performance that approaches the centralized accuracy level.  We also conducted a new experiment to compute the speed-up when varying the number of clients from two to five and keeping fixed the dataset size to 31.5 K samples. This in particular could not be done considering the second scenario we described in the original version of the manuscript, where the number of data samples is equal for each client (i.e., 6.35 K) and increases according with the number of participants to the training process. The obtained results from this new experiment demonstrate an increment of the FL speed-up ranging from the 27% with two clients up to the 35% with five. With regards to the accuracy, since in this new experiment we fixed the dataset size, it remained almost constant, thus resulting in an overall accuracy loss of about the 0.26%. The 90.25% and 94.375% are the accuracies related to the FL and shown in Figure 10b. Here our goal was to demonstrate how the accuracy performance of FL changes when varying the number of clients. All the above explained aspects have been reported in lines (331-338), (347-351), (354-356), and (381-389)  at pages 11,12,13  and in Figures 9,11.

Q2.2:

The accuracy between federated and centralized approaches is very close. How can this be justified when the overhead and complexity of the federated approach is considered?

A2.2:

According to the context, the use of a Federated Learning approach can be less or more effective. In general, the use of this approach is a valid choice when working in application scenarios where a large amount of data is required. In such a context, Federated Learning allows to split the workload among a set of participants while preserving data privacy, but most importantly maintaining comparable performance when compared with a centralized approach. Of course, in some cases the presence of too large overheads or the overall complexity during the implementation of this technique could make a centralized approach more efficient.  In this sense, Federated Learning should be seen not as a replacement of more traditional techniques, but as a valid alternative when specific requirements in terms of hardware, software and application context are satisfied. In the revised version of the manuscripts we added these considerations in lines (396-398) at page 13.

Q2.3:

Varying the number of clients from 2 to 5 does not provide enough details with respect to big data. The motivation is that big data is getting overwhelmingly large, yet the analysis is done with up to 5 clients. A more structured experiment is needed to evaluate the outcomes of FL (in this case the impact of number of clients on accuracy)

A2.3:

Emulating a Federated scenario is quite complex. This problem could be solved by simulating a large number of clients, however our goal in this work is to propose a real implementation of a federated approach, considering a realistic PoC where the use of this approach can be beneficial. Moreover, the number of clients participating to the training process  is strongly dependent on the application context. For example, if we consider a scenario where the clients are represented by organizations, it is quite common to have a number of participants in the same range of the one proposed in the manuscript. This aspect has been added in the revised version of the manuscript in lines (256-259) at page 8.

Q2.4:

Proofreading is required. There are several sentences that are hard to read or could be written better. For instance, “actuators has produced (and still produce)” can be replaced with “have been producing”, “making more difficult its organization”, etc.

A2.4:  

We fixed the sentences as you suggested, and carefully proofread the manuscript in order to remove typos and hard to read sentences. 

Reviewer 3 Report

The paper “Leveraging Stack4Things for Federated Learning in Intelligent Cyber Physical Systems” proposes an implementation of a federated learning algorithm on a custom cloud platform (Stack4Things). The introductory section of it debates the opportunity of using a distributed learning algorithm for IoT based applications, with emphasis on efficiency and preservation of data privacy. The discussion extends on a very brief related work section, which presents a few examples of state of the art federated learning methods.  The authors use this opportunity to reveal some weaknesses of the existing concepts and to motivate their work. The next sections present the Stack4Things architecture and an implementation of a federated learning approach on this architecture. The implementation is then validated on a testing environment involving five clients as laptops, RaspberryPI and Jetson Nano devices.

The paper contains an interesting approach for increasing the efficiency of the training process of a machine leaning algorithm in a Cloud + IoT specific environment. However, it mixes several ideas in an inconsistent manner, which makes the identification of authors contribution more difficult.

  • The authors use an existing federated learning algorithm based on the well-known weights averaging method. No contributions were added to improve the result of the algorithm. Therefore, the results presented in the experiments section does not prove much related with the authors’ work.
  • There is no discussion on the influence of the clients’ heterogeneity on the algorithm efficiency (e.g. as the server must wait for all clients to end all the epochs and rounds – on the line 12 of the Algorithm 1 – the overall performance of the system will depend on the weakest client).
  • The use case selected for validation is not very convincing for the purpose of the system as the training process does not depend on the local collected data. In this particular case, will be more appropriate to run a centralized training on a very powerful machine (only once, in the installation phase) and to use the distributed devices only to run the pretrained algorithm.
  • The results presented in the paper heavily depends on the hardware resources available on the server side (centralized approach used for comparison) and on the client side. These results are slightly irrelevant as they cannot be generalized when other kind of resources are involved, or when other kind of application is considered.
  • The related work section is very poor, and it is not entirely relevant for the authors’ work.

Author Response

Q3.1:

The authors use an existing federated learning algorithm based on the well-known weights averaging method. No contributions were added to improve the result of the algorithm. Therefore, the results presented in the experiments section does not prove much related with the authors’ work.

A3.1:

As we explained in the Introduction section, the main contribution of our work is related to the extension of the Stack4Things platform developed in our department. In this sense, the use of this platform allows a fast deployment of Federated Learning schemes on distributed heterogeneous clients. Compared with the platforms today available (e.g., PySift, LEAF, Flower, and TensorFlow Federated)  Stack4Things allows us to solve the problem related to the client addressing (even if behind NAT) without caring about their hardware and software configurations. We exploited this functionality to extend the framework features by implementing an AI engine both on the client and gateway sides thus enabling Stack4Things to perform training and inference tasks. With the experimental results, we wanted to  demonstrate the feasibility of Stack4Things as an effective framework for  the training of complex deep learning algorithms. This aspect has been better clarified in the introduction section in lines (64-66) at page 2.

Q3.2:

There is no discussion on the influence of the clients’ heterogeneity on the algorithm efficiency (e.g. as the server must wait for all clients to end all the epochs and rounds – on the line 12 of the Algorithm 1 – the overall performance of the system will depend on the weakest client). 

A3.2:

Thank you for your suggestion. Indeed, the clients heterogeneity is a very challenging problem that can heavily affect the Federated Learning performance. In this sense, (as you mentioned) the weakest client will define the performance of the entire system. In the revised version, we partially addressed this problem by adding a timeout mechanism. Specifically when the Cloud waits for the clients models, It triggers a timeout mechanism after which the connection with those clients not satisfying this time requirement will be cut. By doing so, we are able to mitigate this problem also addressing those conditions where one or more clients have a fault, thus causing an undefined waiting time on the Cloud side. Another solution consists in the use of more advanced techniques like quantization or compression that allow to reduce a deep learning model complexity, thus making its training process suitable even for those clients with hardware constraints.  These aspects have been explained in the revised version of the manuscript in lines (214-220), (362-372) at pages 6 and 13.

Q3.3:

The use case selected for validation is not very convincing for the purpose of the system as the training process does not depend on the local collected data. In this particular case, will be more appropriate to run a centralized training on a very powerful machine (only once, in the installation phase) and to use the distributed devices only to run the pretrained algorithm.

A3.3:

The solution you propose works as long as the data is not private. Considering a scenario where the parts do not want to share their data, this approach can’t be used. Federated Learning solves this privacy problem allowing the training of complex deep learning models without the need of sharing the data. Moreover, the use of Federated Learning allows also the implementation of continuous learning mechanisms providing the “virtual” access to a very large number of training samples that can be exploited to periodically update the clients models, thus improving the overall performance of a task. This last aspect in particular has been added in the revised version of the manuscript and can be found in lines (275-279) at page 9.

Q3.4:

The results presented in the paper heavily depends on the hardware resources available on the server side (centralized approach used for comparison) and on the client side. These results are slightly irrelevant as they cannot be generalized when other kind of resources are involved, or when other kind of application is considered.

A3.4: 

We agree that the results depend on multiple factors like the hardware resources and on the application. However, it has been widely demonstrated the effectiveness of Federated Learning over a centralized approach in a large number of scenarios.  We built an experimental setup with realistic hardware and software resources, and we considered a dataset with a large number of samples to best emulate the Federated Learning scenario. In this sense, even if the results change according to the experimental setting, the Federated Learning capability to create a collaborative model through the clients workload split (while saving privacy) is generalizable and applicable in many different contexts. The experimental results we reported in the manuscript further prove this effectiveness. These aspects have been discussed in lines (256-259) and (289-291) at pages 8 and 10. 

Q3.5:

The related work section is very poor, and it is not entirely relevant for the authors’ work.

A3.5:

In the related work, we reported the definition of Federated Learning and described some application scenarios where the use of this approach can be beneficial. The main contribution of our work consists in the implementation of a Federated Learning framework leveraging the functionalities of the Stack4Things platform. In this sense, the related works section reports a comparison with the main Federated Learning frameworks such as: TensorFlow Federated, PySift, LEAF and Flower where we put in evidence the differences and the advantages of using our platform. In the revised version of the manuscript, we added more works to enrich the related work section and they can be found in lines (113-123) at page 3.

Round 2

Reviewer 3 Report

The updated version of the manuscript “Leveraging Stack4Things for Federated Learning in Intelligent Cyber Physical Systems” represents an improvement on what was initially reviewed. The authors have addressed most of the comments, and they successfully removed most of the signaled errors.

The literature review coverage has been extended in this new version with benefits on understanding the addressed problem.

The purpose of the work is better delimited now in the Introduction section. Some confusing affirmations where clarified, and the relevance of the work was correctly placed in a restricted context of Stack4Things Cloud based platform.

The serious flow caused by insufficient analysis of the relation between the method’s performance and the clients heterogenic characteristics was corrected by the authors. A convincing solution, which at least partially covers this problem, was developed based on a timeout mechanism described in section discussing the implemented federated approach.

The authors answer to the observation “The use case selected for validation is not very convincing for the purpose of the system as the training process does not depend on the local collected data” is not completely satisfactory. Here the problem relates with the relevance for this use case in validating the presented work, not in the problematic described by this use case itself.

In my opinion, this version of the manuscript can be considered for publication after some minor adjustments.

Author Response

Q3.3 (second round):

The authors answer to the observation “The use case selected for validation is not very convincing for the purpose of the system as the training process does not depend on the local collected data” is not completely satisfactory. Here the problem relates with the relevance for this use case in validating the presented work, not in the problematic described by this use case itself. In my opinion, this version of the manuscript can be considered for publication after some minor adjustments.

A3.3 (second round): Thank you for your suggestion, in the previous version of the manuscript we did not clarify this aspect. Each camera has its own specific framing and setting (i.e., the positioning, the exposure parameter, the  shutter length, etc.), which make the captured images very different from the others. Since cameras images are very different, the training process strongly depends on their local data, which makes the use of more sources of information a key element to improve the overall performance of the system. In such a context, the use of a Federated Learning approach is a valid solution to merge these information while abating the training time and saving privacy. All these aspects have been added in the revised version of the manuscript and can be found highlighted in blue in lines (266-268), (274-276) at pages 9.